# Halting the Metabolic Complications of Antipsychotic Medication in Patients with a First Episode of Psychosis: How Far Can We Go with the Mediterranean Diet? A Pilot Study

**DOI:** 10.3390/nu14235012

**Published:** 2022-11-25

**Authors:** Savina Ntalkitsi, Dimitris Efthymiou, Vasilios Bozikas, Emilia Vassilopoulou

**Affiliations:** 1Department of Nutritional Sciences and Dietetics, International Hellenic University, 57400 Thessaloniki, Greece; 2II. Department of Psychiatry, Division of Neurosciences, School of Medicine, Aristotle University of Thessaloniki, 54124 Thessaloniki, Greece

**Keywords:** first-episode psychosis, nutritional intervention, Mediterranean diet, MedDietScore, body weight, body fat, muscle mass, dietary intake, cholesterol, LDL, HDL, triglycerides, blood glucose, urea

## Abstract

Patients with first-episode psychosis (FEP) often adopt unhealthy dietary patterns, with a risk of weight gain and metabolic and cardiovascular disease. In 21 FEP patients receiving nutritional intervention based on the Mediterranean diet (MedDiet), we explored differences in anthropometric and biometric parameters, according to their antipsychotic (AP) medication: AP1, associated with a lower risk, or AP2, associated with a higher risk of weight gain and metabolic complications. The blood biochemical profile was recorded before and after dietary intervention, and dietary habits and body composition were monitored for six months. Following intervention, all of the patients recorded significant increases in the consumption of fruit and vegetables and decreases in red meat and poultry consumption, with closer adherence to the MedDiet and a reduction in the daily intake of calories, carbohydrates, and sodium. Vegetable consumption and energy, protein, and carbohydrate intake were lower in AP1 patients than in AP2 patients. There was no significant weight gain overall. A reduction was demonstrated in total and LDL cholesterol, sodium, urea, and iron (lower in AP1 patients). It was evident that AP medication affected blood levels of lipids, urea, and iron of FEP patients, but MedDiet nutritional intervention led to a significant improvement in their eating habits, with a restriction in weight gain and a decrease in blood sodium and urea.

## 1. Introduction

Patients with a first episode of psychosis (FEP), meaning that the person is experiencing a psychotic episode for the first time [1], are observed to adopt less healthy dietary habits compared with the general population [2,3]. They consume more calories [4], refined carbohydrates and saturated fat, and less fiber, omega-3 and omega-6 fatty acids, vegetables and fruits, and smaller quantities of some vitamins (B12, B6, C, folate) and minerals, including zinc (Zn) and selenium (Se) [2,5]. One study, in which dietary intake was evaluated using the validated Australian Recommended Food Score (ARFS) assessment tool, reported that people with FEP recorded a low average score in the quality of their diet, placing them in the category of poorest quality nutrition [4]. In a recent review, although most studies reported that patients with psychosis consumed more calories, the association between total caloric intake and psychosis was unclear [5]. A study on the Mediterranean diet (MedDiet) reported that patients with FEP showed lower adherence to the MedDiet pattern than other subjects [6].

Patients with psychosis tend to suffer from nutrient deficiencies, resulting from inadequate dietary intake or malabsorption. The effect of these deficiencies is bidirectional. Some deficiencies are risk factors for mental disorders and may play a role in the development of psychotic illness and persistent schizophrenia [7]. Folate, B6, and B12 are vitamins considered to be involved in the development of schizophrenia [8]. Several meta-analyses have demonstrated significantly low serum levels of folate [9,10] and vitamin B6 [11] in patients with FEP, but also significant vitamin D deficiency [12]. Lower levels of folate and vitamin D were detected in patients with FEP, but no significant evidence was found for an effect of other vitamins and minerals, including vitamins A, E, and B12, and the minerals zinc (Zn), magnesium (Mg), sodium (Na), potassium (K), calcium (Ca), copper (Cu), chromium (Cr), iron (Fe), manganese (Mn), and selenium (Se) [13].

Compared with the general population, psychiatric patients, particularly those with severe mental illness, have poorer physical health and a much shorter life expectancy, due mainly to premature onset of cardiovascular disease (CVD) [14]. This finding has been linked to the use of antipsychotic drugs (APs) and other aggravating factors (smoking, self-neglect tendencies, unhealthy lifestyle, and low socioeconomic status), which contribute to an increase in the rates of metabolic syndrome, diabetes mellitus (DM), and CVD, with rapid weight gain and obesity, hypertension, and dyslipidemia [15,16,17,18,19,20,21,22].

Central obesity rates are up to three times higher in people with a mental illness than in the general population [23]. Some atypical APs, such as olanzapine, asenapine, clozapine, and risperidone are associated with a higher risk of weight gain and metabolic complications [24,25,26], while others, such as ziprasidone, [27,28], aripiprazole, amisulpride, quetiapine, paliperidone, and ziprasidone, appear to cause fewer metabolic complications [24,25,26].

Some risk factors are inherently non-modifiable (i.e., gender, age, family history, etc.), while others are partially or even completely modifiable through behavioral changes (i.e., obesity, smoking, glucose levels, hypertension, and dyslipidemia) [20]. For this reason, factors such as dietary modification, particularly adopting the MedDiet habits [29], and maintaining a healthy body weight are important for reducing cardiovascular risk and other metabolic complications of psychosis and APs. In patients with FEP, specifically, unhealthy dietary intake appears to be multifactorial, and the contributing factors include: (a) increased appetite due to psychotropic drugs, particularly APs such as olanzapine and clozapine, (b) lack of self-discipline in food consumption, due to cognitive impairment, (c) financial constraints, and (d) lack of motivation [30]. It has recently been reported that poor dietary habits are not due solely to a lack of knowledge of the patients, supporting the need for forms of intervention that will provide more than simple nutritional information [31].

Nutritional intervention for the prevention and treatment of mental illness has received increasing attention in recent years, but intervention programs targeting individuals with FEP have been limited. Lifestyle modification can be a challenge in this group of patients, who are often further hampered by cognitive deficits resulting from the effects of both the disease itself and the AP medication [32,33,34].

Previous efforts towards nutritional intervention have focused on improving body weight, cardiometabolic factors, and mental health. Regarding body weight, a meta-analysis of 13 interventional studies in patients with psychosis showed a small reduction in mean BMI of 0.98 kg/m^2^ [35]. Focusing on cardiometabolic factors, one meta-analysis showed a significant improvement in waist circumference (WC) and in blood levels of triglycerides, fasting glucose, and insulin, while no significant effects were found on BP or blood cholesterol level [36]. In contrast, another meta-analysis found that fasting glucose and triglyceride levels were not significantly affected by dietary intervention [37]. Regarding mental health, a diet rich in omega-3 fatty acids and antioxidants has been documented to be associated with the prevention and improvement of psychiatric symptoms [38,39].

It has been reported that patients with FEP exhibit lowered levels of motivation [40], explaining, in part, the difficulties in achieving the desired dietary intervention outcome [31], but nutritional intervention can have a positive impact on the physical health of these individuals [41], and specifically, individualized dietary advice with weekly shopping tours and cooking groups, in a community program, was shown to be effective in reducing WC and cardiometabolic risk in 30 young people with FEP [42]. A 12-month dietary intervention, which included 24 h recalls, diet history, educational modules, and goal setting in young people with FEP, aged 15–25 years, recorded a 24% reduction in daily energy intake and a 26% reduction in daily Na intake [43], although the improvement in the quality of the diet was not significant, apart from an increase in vegetable intake [43]. A comparative nutritional intervention in 16 young people with FEP, aged 14–25 years, resulted in significantly less weight gain (1.8 kg) in the intervention group, which received health coaching, dietetic support, and supervised exercise prescription over a 12-week period, compared with the control group (7.8 kg), the members of which received standard care involving individual mental health case management and antipsychotic prescription [44].

Finally, a recent review highlighted the beneficial effects of combined dietary and exercise intervention on the physical and mental health of patients with FEP but confirmed that behavioral change in these participants may be difficult to achieve, due to factors related to both the psychosis and its medication [45].

In this 6-month nutritional intervention study based on the MedDiet, we aimed to investigate in patients with FEP receiving two categories of AP medication: (a) the level of improvement in their dietary habits in terms of adherence to the MedDiet, (b) the effects of the dietary intervention on their anthropometric measurements, and (c) the effects of the dietary intervention on selected biochemical markers.

## 2. Materials and Methods

### 2.1. Sample

The study involved 21 outpatients with a diagnosis of FEP, as determined after psychiatric and neurological evaluation by psychiatrists from the 2nd University Psychiatric Clinic of the Psychiatric Hospital of Thessaloniki, according to the International Classification of Disorders (ICD10) [46]. Patients were included who were diagnosed with schizophrenia (F20), chronic delusional disorder (F22), unspecified non-organic psychosis (F29), and bipolar affective disorder with a recent manic episode with psychotic symptoms (F31.2). The clinical history and medication data were retrieved from the patients’ hospital medical records. Based on the type of AP treatment, the patients were divided into two groups: (a) AP1, taking those APs associated with a lower risk of weight gain and metabolic complications (i.e., aripiprazole, amisulpride, quetiapine, paliperidone, and ziprasidone), and (b) AP2, associated with a greater risk of weight gain and metabolic complications (i.e., olanzapine, asenapine, clozapine, and risperidone) [24,25,26]. In this study, we included only patients who had experienced a recent (within six months) psychotic episode, were receiving medication, and had an alteration in any metabolic factor (body weight, blood glucose, blood lipids, or blood pressure).

The patients were informed about the study through oral communication and an information form, and all of them provided their signed consent for participation in the study.

The study was approved by the scientific committee of the Psychiatric Hospital of Thessaloniki Psychiatric Hospital of Thessaloniki and by the Bioethics Committee of the School of Medicine of the Aristotle University of Thessaloniki (AUTH) (code number: 3.303/3.22/12/2020), in full compliance with the International Code of Medical Ethics of the World Medical Association.

### 2.2. Assessment of Current Dietary Intake

During the first interview of each participant with a dietician, a dietary habits questionnaire was completed regarding their dietary history (Appendix A).

Adherence to the MedDiet was assessed with the Mediterranean diet score (MedDietScore) questionnaire [47,48]. Briefly, the MedDietScore records the weekly frequency of consumption of potatoes, unrefined grains, legumes, vegetables, fruit, red meat, poultry, fish, dairy, olive oil, and alcoholic beverages in 6 categories ranging from rarely to daily. The total score is calculated from the sum of the scores obtained from all food groups. For the purposes of this study, the monthly MedDietScore was calculated.

During the intervention, random 24 h dietary intake recalls were obtained by personal interview via telephone twice a month. The interview was structured to help the respondent recall all of the the food and drink consumed the previous day; thus, the type and quantity of food consumed, and the way it was prepared, were recorded accurately.

The nutrient content of the 24 h dietary intake recalls was analyzed using the ESHA Food Processor database (Version 7.30, ESHA Research, Salem, OR, USA) [49].

### 2.3. Nutritional Intervention Process

The nutritional intervention consisted of: (a) a standard weekly diet plan (1500 Kcal for women and 2000 Kcal for men) with meal options for each of the 5 daily meals, based on the MedDiet; (b) printed nutritional education tools based on the MedDiet, including the Mediterranean food pyramid [50,51] and the Healthy Eating Plate [52]; (c) a food portion poster [53] aiming at better control over food portions; (d) individualized nutritional advice aimed at achieving and/or maintaining the patients’ BMI at normal limits and/or improving the hematological/biochemical indicators; (e) clear instructions for preparing the meals contained in the diet plan.

### 2.4. Anthropometric Measurements

Each participant underwent measurement of body weight (kg), body fat (%), muscle mass (kg), muscle quality (MQ), bone mass (kg), visceral fat (LV), resting basal metabolic rate (Kcal/day), metabolic age (years), and body water (%) at baseline and at each monthly session. The measurements were made with the Tanita scale RD-545 model (“RD-545-Connected smart scale | Tanita Official Store”, n.d.) to the nearest 0.1 kg, based on the bioelectrical impedance (BIA) method. The measurements were made in the morning hours, following overnight fasting, and the participants were barefoot and wearing light clothing.

The height was measured at baseline with a height meter (Leicester Height Measure, Invicta Plastics Ltd., Oadby, UK), to the nearest 0.1 cm, with the participants barefoot and with their shoulders in a relaxed position, their arms hanging freely, and their heads in the horizontal plane.

BMI was calculated using current weight and height as (weight (kg) to height squared (m^2^) (kg/m^2^)).

### 2.5. Hematological/Biochemical Assessment

A venous blood sample was collected from each participant as part of the routine monitoring procedure at the beginning (T0) and end of the nutritional intervention (T1).

Blood was withdrawn after overnight fasting and the samples were analyzed using an automatic analyzer (Toshiba TBA 120FR; Toshiba Medical Systems Co., Ltd., Tokyo, Japan) under standard conditions on the hospital premises. The following blood analysis was conducted: red blood cells (RBC) (M/μL), white blood cells (WBC) (K/μL), platelets (PLT) (K/μL), iron (Fe^2+^) (μg/dL), ferritin (ng/mL), vitamin B12 (pg/mL), urea (mg/dL), uric acid (mg/dL), creatinine (mg/dL), serum oxaloacetate transaminase (SGOT) (IU/L), serum pyruvate transaminase (SGPT) (IU/L), lactic dehydrogenase (LDH) (IU/L), γ-glutamyltransferase (γ-GT) (IU/L), C-reactive protein (CRP) (mg/dL), total cholesterol (mg/dL), high density lipoprotein (HDL) (mg/dL), low density lipoprotein (LDL) (mg/dL), triglycerides (mg/dL), blood glucose (mg/dL), serum Na^+^ (mEq/L), and serum K^+^ (mEq/L).

### 2.6. Statistical Analysis

The data were analyzed using the Statistical Package for the Social Sciences (SPSS) and are presented as mean (±standard deviation, SD) or median (min, max). The Kolmogorov–Smirnov normality test (with Lilliefors correction) was used to test for normal distribution. Comparisons before (T0) and after (T1) the intervention and the two groups (AP1 and AP2) of patients were performed with the non-parametric Wilcoxon test (for quantitative variables) and Pearson’s χ^2^ test (for categorical variables). Differences with a *p*-value less than or equal to 0.05 (*p* ≤ 0.05) were considered significant. The effect size was calculated with the Cohen’s d for change in the adherence to the MedDiet.

## 3. Results

### 3.1. Demographic Characteristics

A total of 21 patients (61.9% men) participated in the present study, with a mean age of 35.9 ± 10.2 years. The characteristics of the participants are shown in Table 1. The patients were classified into two groups based on their medication, AP1 or AP2, with 11 patients in the AP1 group (52.4%, 63.6% male) and 10 patients (47.6%, 61.9% male) in the AP2 group (*p* > 0.05). The AP1 patients had a mean age of 30.3 ± 7.44 years, while the mean age of the AP2 group was 42.1 ± 9.43 years (*p* > 0.05). Overall, 10 patients were diagnosed with schizophrenia (F20), 1 with delusional psychosis, 9 with unspecified non-organic psychosis (F29), and 1 with bipolar disorder with psychotic features. No other chronic disease was present in 18 (85.7%) of the patients, while 1 patient in the AP1, and 2 in the AP2 group (14.3%) suffered from tachycardia, asthma, and an antibiotic allergy (*p* > 0.05). Four patients (36.4%) in the AP1 group and 3 (30%) in the AP2 group reported a family history of mental illness (*p* > 0.05) (Table 1).

### 3.2. Anthropometric Measurements

As shown in Table 2, at T0, patients from both groups were overweight with a mean BMI of 26.6 (±4.89), with no significant difference between the groups. The overall visceral fat rate was within the normal range (7.55 ± 3.36), with no difference between the two groups. Following intervention (T1) no significant changes in body weight were observed, as for all patients, the mean body weight was 80.9 ± 18.2 kg at T0, and 80.3 ± 18.1 kg at T1 (*p* = 0.85). There were no significant differences between the two groups in the rest of the anthropometric parameters and no changes in either group from T0 to T1 (Table 2).

### 3.3. Modification in Eating Habits

Regarding dietary habits and adherence to the MedDiet, changes were recorded over the course of the study, both between groups and within each group, as shown in Table 3 depicting the dietary habits and Table 4 depicting the specific nutrient intake.

In the AP1 group, the weekly frequency of vegetable consumption was 1 (1.00, 4.00) (1–6 servings) at T0, and it increased to 3 (3.00, 4.00) (13–20 servings) at T1 (*p* = 0.02). In the AP2 group, the weekly frequency of vegetable consumption was 2 (1.00, 3.00) (7–12 servings) at T0, and it increased to 4 (3.00, 5.00) (21–32) at T1 (*p* < 0.001). All of the study patients increased their mean weekly frequency of vegetable consumption significantly, from 1 (1.00, 4.00) (1–6 servings) at T0 to 3 (1.00, 5.00) (13–20 servings) at T1 (*p* < 0.01), but group AP2 patients consumed, at all times, significantly greater amounts of vegetables (*p* = 0.05) (Table 3).

A mean significant increase in the weekly frequency of fruit consumption was observed in all of the patients, from 5–8 servings (2 (0.00, 5.00)) at T0 to (9–15 servings) (3 (1.00, 5.00)) at T1 (*p* < 0.01). In the AP2 group, the weekly frequency of fruit consumption increased significantly, from 5–8 servings (2 (0.00, 5.00)) at T0 to (16–21 servings) (4 (1.00, 5.00)) at T1 (*p* = 0.02) (Table 3).

Overall, the mean frequency of red meat consumption decreased between T0 and T1 (*p* < 0.01). The AP2 patients reduced their meat intake from 4–5 servings (3 (1.00, 5.00)) at T0 to 2–3 servings (4 (3.00, 5.00)) at T1 (*p* = 0.04). Similarly, the mean poultry consumption decreased between T0 and T1 in both groups (*p* = 0.05) (Table 3).

The overall adherence to the MedDiet, as measured with the MedDietScore index, showed significant increases between T0 and T1 (*p* < 0.01). The AP1 group recorded a mean MedDietScore of 33 (26.0, 37.0) at T0, which increased to 37 (29.0, 43.0) at T1 (*p* = 0.02). The MedDietScore index in the AP2 group, increased from 31 (22.0, 35.0) at T0 to 39 (34.0, 49.0) at T1 (*p* < 0.001). No significant differences were observed in the consumption of the rest of the food groups, either between T0 and T1 or between AP1 and AP2 (*p* > 0.05) (Table 3). The effect based on the change on the adherence to the MedDiet among the groups was large (Cohen’s d = 0.8).

Analysis of the 24 h dietary recall, as presented in Table 4, showed that at T1 the AP1 group had a mean daily intake of 1220 ± 287 Kcal, while that of the AP2 group was 1650 ± 553 Kcal (*p* = 0.05). Overall, the patients in both groups reduced their mean daily energy intake from 1730 ± 504 Kcal to 1430 ± 477 Kcal (*p* = 0.05) (Table 4).

At T1, the AP1 group had a mean daily protein intake of 51.6 ± 22.3 g, and in the AP2 group this was 77.7 ± 27.6 g per day (*p* = 0.03). The mean daily carbohydrate intake of the patients in the AP1 group decreased from 177 ± 80.4 g at T0 to 118 ± 52.6 g at T1 (*p* = 0.04). In comparison, at T1 the AP2 group had a mean daily carbohydrate intake of 175 ± 76.6 g (*p* = 0.05). The carbohydrate intake of all of the patients decreased from a mean of 193 ± 71.7 g to 145 ± 70 g per day (*p* = 0.03) (Table 4).

The daily dietary intake of vitamin E of the AP1 group was 10.6 ± 7.82 mg at T0 and 6.56 ± 4.24 mg at T1 (*p* = 0.05). At T0, the dietary intake of vitamin B3 was significantly lower in the AP1 group than in the AP2 group, with mean values of 11.8 ± 3.88 mg and 19.3 ± 6.76 mg, respectively (*p* = 0.01) (Table 4).

Finally, the daily dietary Na intake was reduced in all of the patients, from 2010 ± 1040 mg at T0 to 1290 ± 778 mg at T1 (*p* = 0.01). This reduction was most pronounced in the AP2 group, from 2060 ± 972 mg at T0 to 1190 ± 653 mg at T1 (*p* = 0.03) (Table 4).

The rest of the macronutrients and micronutrients consumed daily by the patients showed no significant change between T0 and T1, or differences between the patient groups (Table A1).

### 3.4. Hematological and Biochemical Indices

The main findings from analysis of the blood of the patients over the study period are presented in Table 5. The laboratory data showed several significant differences between the two groups at both T0 and T1, and fewer differences between the two time periods within each group and for all of the patients.

Specifically, at T0, the mean serum Fe^2+^ value was 60.8± 16.1 μg/dL in the AP1 group and 86.1± 32.1 μg/dL in the AP2 group (*p* = 0.02). At T1, the mean serum Fe^2+^ value was 53.9 ± 19.7 μg/dL in the AP1 group and 109 ± 29.8 μg/dL in the AP2 group (*p* < 0.001). Between T0 and T1 no significant change in serum Fe^2+^ values was observed for each group separately for all of the patients (Table 5).

The mean total blood cholesterol at T0 in the AP1 group was 179 ± 41.6 mg/dL, and in the AP2 group it was 220 ± 41.9 mg/dL (*p* = 0.03). The total cholesterol showed no significant change after dietary intervention in either group or overall (Table 5).

The mean LDL cholesterol at T0 was 130 ± 102 mg/dL in the AP1 group and it was 132 ± 34.5 mg/dL) in the AP2 group (*p* = 0.05). At T1, the mean LDL cholesterol was 124 mg/dL (±90.5) in the AP1 group and 143 mg/dL (±37.3) in the AP2 group, and there were no significant changes after intervention (Table 5).

The mean blood urea at T0 in the AP1 group was 23.3 ± 6.36 mg/dL, which was lower than that in the AP2 group at 29.6 ± 6.90 mg/dL (*p* = 0.05), decreasing to 24.8 ± 3.46 at T1. The changes in mean blood urea between T0 and T1 were not significant overall (Table 5).

The mean fasting glucose at T1 was 97.5 ± 7.35 mg/dL in the AP1 group and 90.8 ± 6.11 mg/dL in the AP2 group (*p* = 0.05). The differences in mean fasting glucose between T0 and T1 were not significant (Table 5).

The mean serum Na^+^ concentration at T0 was 140 ± 1.29 mEq/L in the AP1 group, decreasing to 139 ± 1.51 mEq/L at T1 (*p* < 0.01). Similarly, the mean serum Na^+^ concentration of the total sample of patients was 140 ± 1.25 mEq/L at T0 and this decreased to 139 ± 1.8 mEq/L at T1 (*p* < 0.001) (Table 5).

No significant differences between the two groups, or between measurements at T0 and T1 were observed in the other hematological and biochemical indices investigated (Table A2).

## 4. Discussion

A six-month nutritional intervention based on the MedDiet in overweight patients with FEP resulted in several significant differences in the patients’ eating habits and in their nutrient intake, with changes in some biochemical indices, the most significant of which were observed in AP2 patients receiving APs mainly linked with greater weight gain and metabolic complications. Although no significant changes in the anthropometric measurements were observed following intervention, it is of note that there was no weight gain over the study period in either group despite the continuous use of APs.

Regarding their dietary habits over the course of the study, all of the patients increased the frequency of vegetable consumption, with the AP2 group increasing the frequency more than the AP1 group. An increase in vegetable consumption following intervention in patients with FEP was also noted in a previous study where no change was observed in the other food groups [43]. The mean frequency of fruit consumption also increased in the AP2 group, leading to an increase in the intake vitamins, minerals, and fibers, with a corresponding reduction in the risk of nutritional deficiencies, which are often experienced by these patients [5].

In addition, the frequency of red meat and poultry consumption was reduced in the AP2 group, leading to a reduction in the intake of saturated fat, which tends to be high in FEP patients, increasing their cardiometabolic risk [20]. In our study, the dietary intervention was for 6 months, but if dietary modification continued, in the long term, this reduction in saturated fat intake could minimize the dyslipidemias and cardiovascular risk linked with use of the AP2 drugs [18].

Overall, adherence to the MedDiet increased significantly over the course of the intervention [47,48]. The AP2 patients in particular recorded a greater increase in their MedDietScore as they changed their eating habits in the four food groups (vegetables, fruit, red meat, and poultry), compared with one (vegetables) in the AP1 patients. In general, the AP2 group followed the dietary advice more faithfully than the AP1 group, and this could benefit them in the long term [18], as it has been documented that patients taking AP2 drugs are more prone to metabolic disorders [17,24,25,26] and are therefore more likely to need medication at some time to treat DM, dyslipidemia, and CVD [18]. The MedDiet could be an interventive measure to prevent or ameliorate cardiometabolic complications in patients taking APs [54,55]. Based on our results, the dietary habits of FEP patients can be improved and healthier eating patterns through dietary intervention and monitoring can result in long-term improvements in their metabolic health.

According to the regular 24 h dietary intake recall, a significant mean reduction in average calorie intake was observed over the course of the study, by 300 Kcal per day; a similar scale reduction was observed in a previous intervention study [43]. At T1, the AP2 patients consumed more calories, through protein and carbohydrates, than the AP1 patients; calorie intake contributes to the greater gain and difficulty in losing body weight, which has been observed in patients taking AP2s [24,25,26]. In addition, the AP1 patients significantly reduced their average daily carbohydrate intake over the course of the study, which was desirable as patients with FEP tend to consume more carbohydrates than the general population [5]. All of the study patients, in addition to reducing their carbohydrate intake, showed a tendency to decrease their intake of sugars, total fat, saturated fat, polyunsaturated fat, trans fat, and dietary cholesterol over the 6 months; in the longer term such dietary modifications could contribute to the reduction of inflammatory and cardiovascular risk.

Regarding vitamins, overall, the patients’ vitamin requirements, with the exception of vitamin E, B5, and folate, were met throughout the intervention based on the DRIs [56]. The dietary intervention did not lead to a change in vitamin intake (vitamins A, B1, B2, B3, B5, B6, B12, folate, vitamins C, D, and E), and no differences were observed between the groups, with the exception of vitamin B3 which was taken in larger quantity by AP2 patients at T0, meeting their needs based on the DRIs [56], compared with AP1 patients. A low intake of folic acid by patients with a severe mental illness has been reported in a recent review [5], but there are no clear data on the intake of the other vitamins in these patients [57]. Regarding minerals, the observed reduction in the daily dietary intake of Na by the AP2 patients was significant, as at T0 it was higher than the DRIs and at T1 it was within the recommended range [56]. In the total sample of patients, the reduction in daily Na intake was greater than that recorded in a previous intervention study [43], and it could contribute to a reduction in hypertension and cardiovascular risk [58]. Finally, the needs of the study patients for Fe, Ca, K, Zn, and Mg were not met during the intervention, which is consistent with previous findings, where the nutritional deficiency in mineral elements is reported to be mainly due to the dietary choices of people with a severe mental illness [5].

Regarding the anthropometric data, no significant difference was observed between the two groups of FEP patients. Although the mean BMI in both groups of patients showed that they were overweight at the start of the intervention, only in the AP1 group was there a tendency towards a decrease in the mean weight after intervention. Despite the observation that the AP2 group engaged in several healthier eating habits, they showed a tendency towards an increase in body weight, but at a non-significant level, and the same pattern was observed for the body and visceral fat. This may indicate that the AP1 patients were able to lose body weight and fat more easily with the intervention, while the AP2 patients tended to gain weight and fat, even when complying with dietary intervention, reflecting their documented difficulty in weight loss and subsequent increased risk of gaining body fat [24,25,26]. It appears that nutritional intervention programs show greater effectiveness in body weight and fat control in AP1 patients, thus helping to reduce their cardiometabolic risk. Overall, no significant changes were observed, and the intervention led to a prevention of increase in body weight, as in previous studies [36,44]. The potential benefit for the AP2 group is important, as previous studies indicate greater weight gain in this group, putting them at higher risk of serious inflammatory and cardiometabolic complications [6,17]. The fact that the mean weight of the patients remained stable, although they were receiving medication that has been linked to weight gain [24,25,26,59], is a positive outcome of the intervention, since the prevention of weight gain contributes to the prevention of early cardiometabolic disease in this population [43]. An earlier intervention study in patients with FEP, reported a weight gain of 1.8 kg in the intervention group [44] indicating the difficulty of these patients in maintaining their weight, and underlining the significance of our results. The failure of our study patients to lose weight in spite of adherence to the dietary intervention indicates the need for greater discipline in adherence to dietary recommendations, in conjunction with exercise programs [45].

The blood results showed normal levels of serum Fe^2+^ in both groups, but the mean value was lower in the AP1 group, possibly due to a lower dietary intake of heme Fe^2+^, although further investigation is needed on the possible contribution of APs. The total and LDL cholesterol showed normal mean values in the AP1 group, both at T0 and T1, reflecting their lower risk of developing metabolic disease [16,17]. In contrast, the mean values of total and LDL cholesterol in the AP2 group were outside the normal limits, in accordance with data from other studies [17,18]. The mean value of LDL cholesterol tended to decrease in the AP1 group, indicating the possibility of an improvement in their lipid profile solely through changes in their dietary habits. Conversely there appeared to be more difficulty in achieving a change in the blood lipid profile in the AP2 group through dietary modification, which did not appear to counteract the reported tendency for them to increase due to AP2 medication [17,18,60,61]. It is considered that a longer, more stringent, nutritional intervention would be required to achieve an improvement. Overall, no significant changes were observed in either group in relation to total, HDL, and LDL cholesterol and triglycerides, as was noted in a recent study [62], in which, although the BMI of the patients improved, the cardiometabolic factors remained unchanged after the intervention [62]. These results are in agreement with those of several studies, such as those included in a meta-analysis [36] where no significant effects on the cholesterol levels of FEP patients were documented after dietary intervention [36]. In our study, a significant decrease in blood urea was observed in the AP2 patients, in whom a higher value of serum urea had been recorded at T0. This change is possibly due to their reduction in the consumption of red meat and poultry and will contribute to the improvement in the kidney health of these patients, who due to their medication are at risk of a future decline in renal function [60]. The mean level of fasting glucose was higher in the AP1 patients at T1, which is consistent with previous reports that this class of drugs leads to an increase in glucose levels [59], although the values were within normal limits in the two groups of patients, and no significant change in glucose levels during the intervention was observed, as in other studies [37,44,45]. Finally, the serum Na^+^ of the AP1 group and of the whole study sample decreased by 1 mEq/L over the course of the intervention, probably due to the reduction in salt in the dietary intake, thus contributing to a possible reduction in BP, lessening of cardiovascular risk [58], and assisting in the prevention of kidney damage caused by taking AP medication [17].

### Limitations

According to a recent systematic review there is a lack of high-quality studies using validated tools to assess dietary intake in adults with a severe mental illness [63]. In the present study, although validated dietary intake assessment tools were used, specifically the 24 h dietary intake recalls and the MedDietScore questionnaire, difficulties again arose in assessing the accuracy of the reported dietary intake and the dietary habits recorded, as they may be influenced by the memory and/or honesty of the patients, who due to their disease may have experienced a decline in their cognitive functions [32,33,34]. For this reason, it is suggested that other tools should be used to ensure accuracy in assessing the nutritional intake of patients with FEP. Μοreover, the number of participants was relevantly low (*n* = 21). Despite the small sample size, we were able to observe significant alterations in the eating habits among the two groups and some biochemical factors, as the effect size was high. Probably a prolonged intervention could show changes in the anthropometric and other biochemical measurements.

## 5. Conclusions

In conclusion, diet is a particularly important factor for the health of patients with FEP, as they tend to gain weight, both because of their unhealthy eating habits as a result of the disease and because of the AP medication they receive, and they are thus at an increased risk of metabolic complications.

Nutritional intervention, based on the MedDiet, could lead to an improvement in the eating habits of patients with FEP, with a possible reduction in the intake of carbohydrates, Na, and calories, but these changes are influenced by the type of AP medication they are receiving.

Depending on the AP medication, nutritional intervention could result in the improvement of specific biochemical factors, including serum levels of total and LDL cholesterol, urea, and Na^+^, and it offers a potential prevention measure against an increase in body weight and fat in the patients.

The assessment of the patients’ nutritional intake needs to be ensured with additional nutritional tools, as recall is affected by memory–cognitive deficiencies. It is suggested that, in future studies, the patients and their relatives and/or caregivers be asked to record their daily dietary intake in order to achieve a more accurate version of events.

## Figures and Tables

**Table 1 nutrients-14-05012-t001:** Demographic characteristics of patients with a first episode of psychosis (FEP) (*n* = 21) according to the antipsychotic medication prescribed (AP1 or AP2).

Variable	AP1 *	AP2 *	Total	*p*-Value
	11 (52.4%)	10 (47.6%)	21 (100%)	>0.05
**Gender**				
Female	4 (36.4%)	4 (40.0%)	8 (38.1%)	>0.05
Male	7 (63.6%)	6 (60.0%)	13 (61.9%)	>0.05
**Age**				
Mean (SD)	30.3 (7.44)	42.1 (9.43)	35.9 (10.2)	>0.05
**Diagnosis**				
Schizophrenia (F20) Delusional disorder (F22) Unspecified non-organic psychosis (F29) Bipolar disorder with psychotic features (F31.2)	7 (63.6%) 0 (0%) 3 (27.3%) 1 (9.1%)	3 (30%) 1 (10%) 6 (60%) 0 (0%)	10 (47.6%) 1 (4.8%) 9 (42.9%) 1 (4.8%)	>0.05 >0.05 >0.05 >0.05
**Other chronic disease**				
No	10 (90.9%)	8 (80.0%)	18 (85.7%)	>0.05
Yes	1 (9.1%)	2 (20.0%)	3 (14.3%)	>0.05
**Family history**				
No	7 (63.6%)	7 (70.0%)	14 (66.7%)	>0.05
Yes	4 (36.4%)	3 (30.0%)	7 (33.3%)	>0.05

* AP1: antipsychotics with a lower risk of weight gain and metabolic complications (i.e., aripiprazole, amisulpride, quetiapine, paliperidone, and ziprasidone); AP2: antipsychotics with higher risk of weight gain and metabolic complications (i.e., olanzapine, asenapine, clozapine, and risperidone).

**Table 2 nutrients-14-05012-t002:** Anthropometric characteristics of patients with a first episode of psychosis (FEP) (*n* = 21), according to the antipsychotic medication prescribed (AP1 or AP2), before (T0) and after (T1) dietary intervention.

Variable	Period	AP1 * (*n* = 11)	AP2 * (*n* = 10)	Total (*n* = 21)	*p*-Value
**Body Weight (kg)**	Τ0	82.7 (20.8)	78.9 (15.6)	80.9 (18.2)	>0.05
	Τ1	80.3 (19.8)	80.4 (17.1)	80.3 (18.1)	>0.05
	***p*-value**	>0.05	>0.05	0.85	
**Body mass index (BMI) (kg/m^2^)**	Τ0	27.1 (4.44)	26.1 (5.53)	26.6 (4.89)	>0.05
	Τ1	26.4 (4.30)	26.6 (6.14)	26.5 (5.12)	>0.05
	***p*-value**	>0.05	>0.05	0.72	
**Body Fat (%)**	Τ0	29.0 (7.54)	29.2 (8.02)	29.1 (7.57)	>0.05
	Τ1	28.7 (6.86)	29.8 (9.35)	29.2 (7.95)	>0.05
	***p*-value**	>0.05	>0.05	0.99	
**Muscle Mass (kg)**	Τ0	55.5 (13.9)	52.4 (7.44)	54.0 (11.1)	>0.05
	Τ1	54.3 (13.7)	52.7 (7.97)	53.5 (11.1)	>0.05
	***p*-value**	>0.05	>0.05	0.85	
**Muscle Quality (MQ)**	Τ0	48.1 (8.81)	52.6 (8.47)	50.2 (8.75)	>0.05
	Τ1	51.1 (8.32)	53.0 (7.41)	52.0 (7.77)	>0.05
	***p*-value**	>0.05	>0.05	0.59	
**Bone Mass (kg)**	Τ0	2.92 (1.92)	2.8 (0.38)	2.86 (0.56)	>0.05
	Τ1	2.86 (0.676)	2.82 (0.346)	2.84 (0.532)	>0.05
	***p*-value**	>0.05	>0.05	0.89	
**Visceral Fat (LV)**	Τ0	7.77 (4.42)	7.3 (2.74)	7.55 (3.36)	>0.05
	Τ1	7.18 (4.19)	7.55 (3.31)	7.36 (3.71)	>0.05
	***p*-value**	>0.05	>0.05	0.83	
**Basal Metabolic Rhythm (Kcal/day)**	Τ0	1770 (423)	1640 (239)	1710 (346)	>0.05
	Τ1	1740 (415)	1660 (251)	1700 (341)	>0.05
	***p*-value**	>0.05	>0.05	0.89	
**Body Water (%)**	Τ0	53.5 (6.84)	51.0 (6.42)	52.3 (6.60)	>0.05
	Τ1	52.8 (5.21)	50.5 (7.37)	51.7 (6.27)	>0.05
	***p*-value**	>0.05	>0.05	0.93	

The mean (SD) of each variable is presented. * AP1: antipsychotics with a lower risk of weight gain and metabolic complications (i.e., aripiprazole, amisulpride, quetiapine, paliperidone, and ziprasidone); AP2: antipsychotics with a higher risk of weight gain and metabolic complications (i.e., olanzapine, asenapine, clozapine, and risperidone).

**Table 3 nutrients-14-05012-t003:** Dietary habits of patients with a first episode of psychosis (FEP) (*n* = 21) according to the antipsychotic medication prescribed (AP1 or AP2), before (T0) and after (T1) dietary intervention.

Variable	Group	T0	T1	*p*-Value
**Vegetables**	AΡ1 *	1.00 (1.00, 4.00)	3.00 (1.00, 4.00)	0.02
	AΡ2 *	2.00 (1.00, 3.00)	4.00 (3.00, 5.00)	<0.001
	***p*-value**	0.05	0.05	
	Total	1.00 (1.00, 4.00)	3.00 (1.00, 5.00)	<0.001
**Fruits**	AΡ2 *	2.00 (0.00, 5.00)	4.00 (1.00, 5.00)	0.02
	Total	2.00 (0.00, 5.00)	3.00 (1.00, 5.00)	<0.001
**Red Meat**	AΡ2 *	3.00 (1.00, 5.00)	4.00 (3.00, 5.00)	0.04
	Total	4.00 (1.00, 5.00)	4.00 (3.00, 5.00)	<0.001
**Poultry**	AΡ2 *	5.00 (3.00, 5.00)	5.00 (5.00, 5.00)	0.03
	Total	5.00 (3.00, 5.00)	5.00 (4.00, 5.00)	0.05
**MedDiet Score**	AΡ1 *	33.0 (26.0, 37.0)	37.0 (29.0, 43.0)	<0.01
	AΡ2 *	31.0 (22.0, 35.0)	39.0 (34.0, 49.0)	<0.001
	Total	32.0 (22.0, 37.0)	39.0 (29.0, 49.0)	<0.001

The median [min, max] of each variable is presented. MedDiet: Mediterranean diet. * AP1: antipsychotics with a lower risk of weight gain and metabolic complications (i.e., aripiprazole, amisulpride, quetiapine, paliperidone, and ziprasidone); AP2: antipsychotics with a higher risk of weight gain and metabolic complications (i.e., olanzapine, asenapine, clozapine, and risperidone).

**Table 4 nutrients-14-05012-t004:** Intake of specific nutritional components of patients with a first episode of psychosis (FEP) (*n* = 21), according to the antipsychotic medication prescribed (AP1 or AP2), before (T0) and after (T1) dietary intervention.

Variable	Group	T0	T1	*p*-Value
**Energy (Kcal)**	AP1 *		1220 (287)	
	AP2 *		1650 (553)	
	***p*-value**		0.05	
	Total	1730 (504)	1430 (477)	0.05
**Proteins (g)**	AΡ1 *		51.6 (22.3)	
	AΡ2 *		77.7 (27.6)	
	***p*-value**		0.03	
**Carbohydrates (g)**	AΡ1 *	177 (80.4)	118 (52.6)	0.04
	AΡ2 *		175 (76.6)	
	***p*-value**		0.05	
	Total	193 (71.7)	145 (70.0)	0.03
**Vitamin E (mg)**	AΡ1 *	10.6 (7.82)	6.56 (4.24)	0.05
**Vitamin B** **3 (** **mg** **)**	AΡ1 *	11.8 (3.88)		
	AΡ2 *	19.3 (6.76)		
	***p*-value**	0.01		
**Dietary Na (mg)**	AΡ2 *	2060 (972)	1190 (653)	0.03
	Total	2010 (1040)	1290 (778)	0.01

The mean (SD) of each variable is presented. * AP1: antipsychotics with a lower risk of weight gain and metabolic complications (i.e., aripiprazole, amisulpride, quetiapine, paliperidone, and ziprasidone); AP2: antipsychotics with a higher risk of weight gain and metabolic complications (i.e., olanzapine, asenapine, clozapine, and risperidone).

**Table 5 nutrients-14-05012-t005:** Hematological and biochemical indices of patients with a first episode of psychosis (FEP) (*n* = 21), according to the antipsychotic medication prescribed (AP1 or AP2), before (T0) and after (T1) dietary intervention.

Variable	Period	AP1 * (*n* = 11)	AP2 * (*n* = 10)	Total (*n* = 21)	*p*-Value
**Serum Fe^2+^ (μg/dL)**	Τ0	60.8 (16.1)	86.1 (32.1)	72.9 (27.6)	0.02
	Τ1	53.9 (19.7)	109 (29.8)	80.3 (37.4)	<0.001
	***p*-value**	>0.05	>0.05	0.62	
**Total Cholesterol (mg/dL)**	Τ0	179 (41.6)	220 (41.9)	199 (45.9)	0.03
	Τ1	182 (49.6)	220 (34.4)	200 (46.3)	0.05
	***p*-value**	>0.05	>0.05	0.92	
**Low Density Lipoprotein (LDL) (mg/dL)**	Τ0	130 (102)	132 (34.5)	131 (75.4)	0.05
	Τ1	124 (90.5)	143 (37.3)	133 (69.4)	0.05
	***p*-value**	>0.05	>0.05	0.98	
**Urea (mg/dL)**	Τ0	23.3 (6.36)	29.6 (6.90)	26.3 (7.21)	0.05
	Τ1		24.8 (3.46)	24.9 (4.30)	
	***p*-value**		0.05	0.57	
**Glucose (mg/dL)**	Τ0			92.0 (8.75)	
	Τ1	97.5 (7.35)	90.8 (6.11)	94.3 (7.47)	0.05
	***p*-value**			0.31	
**Serum Na^+^ (mEq/L)**	Τ0	140 (1.29)		140 (1.25)	
	Τ1	139 (1.51)		139 (1.80)	
	** *p* ** **-value**	<0.01		<0.001	

The mean (SD) of each variable is presented. The *p*-values of the differences between groups AP1 and AP2, and time periods T0 and T1 are presented. * AP1: antipsychotics with a lower risk of weight gain and metabolic complications (i.e., aripiprazole, amisulpride, quetiapine, paliperidone, and ziprasidone); AP2: antipsychotics with a higher risk of weight gain and metabolic complications (i.e., olanzapine, asenapine, clozapine, and risperidone).

## Data Availability

The data presented in this study are available on request from the corresponding author. The data are not publicly available due to privacy restrictions.

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
