# Peer review of "Halting the Metabolic Complications of Antipsychotic Medication in Patients with a First Episode of Psychosis: How Far Can We Go with the Mediterranean Diet? A Pilot Study"

_nutrients, 2022, doi:10.3390/nu14235012_

Round 1

Reviewer 1 Report

dear colleagues thank you very much for the submission.

the title need to encloud the word pilot study, was it 21 patience this is not a conclusive research.

the introduction has a lot of noise and need to be streamlined and summarized, patience with psychosis are will known to have metabolic disturbance is with or without antipsychotics treatment authors might want to look at meditation naive metabolic syndrome in psychosis.

the introduction failed to established the problem and highlight the important need of interventions to reduce the better than of cardiovascular these is risk and patient with psychosis.

authors need to be familiar with metabolic syndrome with the score and report it boss before and after interventions.

some very important literature has been missed about cardio psychiatry profile and directory interventions including low carb diet and heterogenic diet. these are very important to justify why MeD diet is a feasible approach.

the interventions are very big and need to be explained and further heavy details.

authors need to provide chlorpromazine equivalent dose for each pt and use it in mr (multiple regression).

the inclusion and exclusion a criteria or not very clear and need detailed revision.

sample size calculation need to be included.

effect size is not reported and this is not acceptable. Cohen d or cramer v need to be shown and interpreted and discussed. 

analysis are very basic and ancova/mr need to be considered.

discussion is will return however the language need to be down with 6mo 21 pts authors need to be humble.

I suggest at more limitations of the study including some of the mentioned above e.g. sample limit.

Reviewer 2 Report

The manuscript I reviewed was well written, the project was correctly designed and data are well described, as well as of important value to the scientific community.
However, there are small corrections and changes that could exacerbate the already good content described.

Page 2 - line 54 - All minerals are listed with a chemical symbol in brackets, except for the last that is selenium. Please correct.

Page 6 - Table 2 - Bone Mass - number 1 is subscript, while in all other cases it's not. Please standardize it.

Page 7 - Line 275-280 - The measure unit envisaged by the International System (SI) and therefore scientifically correct for quantifying mass is the kilogram (kg). For small quantities, the use of the gram, a sub-multiple, is permitted. However, the symbol for grams of substance is g and not gr. Please uniform in the text.

Page 8 - Line 303 - Please replace Fe with iron which best represents the parameter in the text and corresponds to what is written in table 5.

Page 9 - Table 5 - Please standardize minerals in extended or symbolic mode (iron and sodium, or Fe2+ and Na+). 

Page 11 - Line 398-407 - Regarding the lack of statistical significance in weight and BMI changes, my suggestion is to adjust and correlate this data by gender. This could allow to ascertain a significance in this parameter.

Page 12 - Line 455 - Limitations - In my opinion, number of patients should be mentioned. In fact, such a small cohort of patients does not give a high power to the study, because some patients have unique characteristics and lack a relative control group.

Page 12 - Line 471 -  Chemical elements described in the manuscript and detected in serum are all coming from salts and minerals. As a consequence their presence in blood serum is in ionic form. Please indicate the charges that the individual ions carry with them and standardize throughout the text (example sodium with + in superscript - Na+).
This will make it possible to better correlate the reference to the mineral, but above all to ascertain the oxidation state of the element in question, a very important parameter for some elements, such as iron which in the 2+ or 3+ oxidative state is involved in different biological processes in the 'body.

Last but not least, despite the fact that the topic has been well introduced and the data effectively discussed, the opinion of this reviewer is that it would be necessary to mention in the introduction and discussion sections an important review recently published and highly related to the topic covered. In fact, this article deals with the correlation between the Mediterranean diet and healthy aging, arguing between the pathological states of cognitive disorders and cardiovascular diseases.
E. Mazza et al. - Mediterranean Diet In Healthy Aging - J Nutr Health Aging. 2021

Round 2

Reviewer 1 Report

thank you for addressing the main concerns